# Influence of Type 2 Diabetes in the Association of *PNPLA3* rs738409 and *TM6SF2* rs58542926 Polymorphisms in NASH Advanced Liver Fibrosis

**DOI:** 10.3390/biomedicines10051015

**Published:** 2022-04-28

**Authors:** Pablo Gabriel-Medina, Roser Ferrer-Costa, Francisco Rodriguez-Frias, Andreea Ciudin, Salvador Augustin, Jesus Rivera-Esteban, Juan M. Pericàs, David Martinez Selva

**Affiliations:** 1Biochemistry Department, Vall d’Hebron University Hospital, 08035 Barcelona, Spain; pgabriel@vhebron.net (P.G.-M.); frarodri@vhebron.net (F.R.-F.); 2Biochemistry and Molecular Biology Department, Universitat Autònoma de Barcelona (UAB), 08193 Barcelona, Spain; jrivera@vhebron.net; 3Clinical Biochemistry Research Team, Vall d’Hebron Institut de Recerca (VHIR), 08035 Barcelona, Spain; 4Centro de Investigación Biomédica en Red de Enfermedades Hepáticas y Digestivas (CIBEREHD), 28029 Madrid, Spain; saugusti@vhebron.net (S.A.); jpericas@vhebron.net (J.M.P.); 5Endocrinology and Nutrition Department, Vall d’Hebron University Hospital, 08035 Barcelona, Spain; aciudin@vhebron.net; 6Diabetes and Metabolism Department, Vall d’Hebron Institut de Recerca (VHIR), Universitat Autònoma de Barcelona (UAB), 08035 Barcelona, Spain; 7Centro de Investigación Biomédica en Red de Diabetes y Enfermedades Metabólicas Asociadas (CIBERDEM), 28029 Madrid, Spain; 8Liver Unit, Internal Medicine Department, Vall d’Hebron Institut de Recerca (VHIR), Vall d’Hebron University Hospital, 08035 Barcelona, Spain

**Keywords:** nonalcoholic steatohepatitis (NASH), advanced fibrosis, *PNPLA3* p.I148M, *TM6SF2* p.E167K, type 2 diabetes (T2D), homeostatic model assessment for insulin resistance (HOMA-IR)

## Abstract

Nonalcoholic steatohepatitis (NASH) is a leading cause of cirrhosis in western countries. Insulin resistance (IR), type 2 diabetes (T2D), and the polymorphisms patatin-like phospholipase domain-containing 3 (*PNPLA3*) rs738409 and transmembrane 6 superfamily member 2 (*TM6SF2*) rs58542926 are independent risk factors of NASH. Nevertheless, little is known about the interaction between IR and T2D with these polymorphisms in the pathogenesis of NASH and the development of advanced fibrosis. Thus, our study aimed to investigate this relationship. This is a cross-sectional study including NASH patients diagnosed by liver biopsy, at the Vall d’Hebron University Hospital. A total of 140 patients were included (93 T2D, 47 non-T2D). T2D (OR = 4.67; 95%CI 2.13–10.20; *p* < 0.001), *PNPLA3* rs738409 and *TM6SF2* rs58542926 polymorphisms (OR = 3.94; 95%CI 1.63–9.54; *p* = 0.002) were independently related with advanced liver fibrosis. T2D increased the risk of advance fibrosis on top of the two polymorphisms (OR = 14.69; 95%CI 3.03–77.35; *p* = 0.001 for *PNPLA3* rs738409 and OR = 11.45; 95%CI 3.16–41.55; *p* < 0.001 for *TM6SF2* rs58542926). In non-T2D patients, the IR (HOMA-IR ≥ 5.2, OR = 14.33; 95%CI 2.14–18.66; *p* = 0.014) increased the risk of advanced fibrosis when the polymorphisms were present (OR = 19.04; 95%CI 1.71–650.84; *p* = 0.042). The T2D and IR status increase the risk of advanced fibrosis in patients with NASH carrying the *PNPLA3* rs738409 and/or *TM6SF2* rs58542926 polymorphisms, respectively.

## 1. Introduction

Non-alcoholic fatty liver disease (NAFLD) is characterized by the accumulation of fat in the hepatocytes and it is considered to be the major cause of chronic liver disease, affecting 25% of general population worldwide [1]. Although simple steatosis is considered a benign clinical state, the progression to non-alcoholic steatohepatitis (NASH) involves necroinflammatory degeneration, cellular balloonization, and may lead to the development of different degrees of fibrosis and eventually cirrhosis and hepatocellular carcinoma [2,3]. Furthermore, advanced stages of fibrosis are associated with an increased risk of overall of cardiovascular and liver-related morbimortality [4,5,6].

NAFLD and metabolic syndrome (MetS) are intimately related entities. Due to their strong bidirectional association, recently the term “NAFLD” was proposed to change into “metabolic-dysfuntion associated fatty liver disease-MAFLD” [7]. Insulin resistance (IR), one of major components of MetS [8], plays a key role in the pathophysiology of NAFLD, by promoting the progression of simple steatosis to liver inflammation and fibrosis [9]. Moreover, the presence of NASH can promote hepatic IR and, therefore, increases the risk of subsequent T2D [10]. The Homeostatic Model Assessment for Insulin Resistance (HOMA-IR) indirectly evaluates hepatic IR and has been widely used in the routine clinical practice and proposed in clinical algorithms of NASH and T2D [11,12,13].

Additionally, T2D is a well-known risk factor for NAFLD [1]. The prevalence of NAFLD among T2D patients raises up to 60–80% and has been consistently shown that T2D acts as a trigger by promoting the progression to NASH and advanced liver fibrosis [14].

In the last decade, several genetic risk factors have been associated with the susceptibility of NAFLD and the development of a progressive disease [15,16]. Among them, single nucleotide polymorphism (SNP) rs738409 of the patatin-like phospholipase domain containing protein 3 (*PNPLA3*) gene [17,18] and SNP rs58542926 of the transmembrane protein involved in molecule transport (*TM6SF2*) gene [19] have been identified in several genome-wide association studies (GWAS) as risk factors for progressive NASH and advanced fibrosis [20]. PNPLA3, or adiponutrin, participates in intracellular lipid remodeling, and the rs738409 polymorphism is associated with increased hepatocellular triglyceride accumulation by restricting substrate access to the enzyme’s catalytic site. Alternatively, TM6SF2 plays a role in VLDL export from liver to serum, where SNP rs58542926 causes impairment in the lipid exportation and is associated with elevations in serum aspartate aminotransferase (AST) and alanine aminotransferase (ALT) [21].

Although the independent roles of T2D, IR, and the several polymorphisms in the pathogenesis and natural history of NASH have been widely investigated [22], to the best of our knowledge, there are no reports addressing the interaction of *PNPLA3* and *TM6SF2* gene variants with T2D and IR in the pathophysiology of NASH and their role in advanced liver fibrosis.

According to this evidence, the present study aimed to evaluate the interaction between the presence of *PNPLA3* and *TM6SF2* gene variants, T2D and IR with the presence of advanced fibrosis in a cohort of patients with NASH.

## 2. Materials and Methods

We performed a cross-sectional study, including consecutive subjects diagnosed with NASH from January 2016 to December 2019 at the Liver Unit of the Vall d’Hebron University Hospital, Barcelona, Spain. The study was conducted according to the Declaration of Helsinki and was approved by the local Ethics Committee (PR(AG)601/2020). DNA, liver, and biochemical samples from patients included in this study were provided by the Vall d’Hebron University Hospital Biobank (PT17/0015/0047), integrated in the Spanish National Biobanks Network, and they were processed following standard operating procedures with the appropriate approval of the Ethical and Scientific Committees. Serum samples were drawn at the same time that liver biopsy was performed, as per protocol. All participants had previously signed the informed consent.

Inclusion criteria: (a) age >18 years; (b) NASH diagnosis by liver biopsy.

Exclusion criteria: (a) high-risk alcohol consumption (>30 g/day for men and >20 g/day for women); (b) other causes of liver disease (viral or autoimmune hepatitis, hereditary hemochromatosis, alcoholic liver disease, liver transplantation, etc.); (c) hepatotoxic drugs; (d) uncontrolled endocrine diseases (hypothyroidism, hypercortisolism, etc.).

Liver histology evaluation according CRN NASH criteria [23]: (a) steatosis was scored 0–3; (b) lobular inflammation was scored 0–3; (c) ballooning (marker of cell injury) was scored 0–2; (d) NASH activity score corresponded to the unweighted sum of the scores for steatosis, lobular inflammation, and ballooning; finally, (e) fibrosis was staged 0–4. Advanced liver fibrosis was defined as the presence of fibrosis grade 3–4 in the histological evaluation.

Metabolic evaluation: T2D was defined according to ADA guidelines [24].

Hepatic IR was indirectly evaluated using the HOMA-IR, based on the formula: fasting glucose (mg/dl)* fasting insulin (μU/) mL/405 [25]. A cut-off ≥ 3.02 has been described as marker of IR in Caucasian population [26]. Patients with T2D on insulin treatment were excluded from the calculation of HOMA-IR.

Genetic analysis: DNA was extracted from serum samples by the MagNa Pure 24 system (Roche Molecular Systems, Inc. (Branchburg, NJ, USA)). The *PNPLA3* rs738409 C > G (I148M) and *TM6SF2* rs58542926 C > T (E167K) SNPs were assessed by allele-specific genotyping techniques with real-time polymerase chain reaction (qPCRr) and fluorescent resonance energy transfer (FRET) specific probes [27] and melting peaks analysis (Light SNiP assay) (TIB MOLBIOL GmbH, Berlin, Germany) on a capillary based LightCycler 2.0^®^ thermocycler (Roche Molecular Systems, Inc. (Branchburg, NJ, USA)).

Statistical analysis: The distribution of data was assessed by the Kolmogorov-Smirnov test. T Student and U Mann-Whitney tests were used to compare quantitative variables, which followed a Gaussian distribution or not, respectively. A chi-squared test was used to compare proportions. The genotype frequencies of the *PNPLA3* and *TM6SF2* polymorphisms were tested for consistency with Hardy-Weinberg equilibrium using exact tests (https://ihg.gsf.de/cgi-bin/hw/hwa1.pl; accessed on 17 January 2022). Allele frequency differences were assessed by a chi-squared test and genotype frequencies were assessed under a dominant genetic model (due to the low number of homozygotes mutant alleles). Logistic regression analysis was performed to study the association of NASH development and degree of liver fibrosis according to clinical, biochemical variables, and *PNPLA3* and *TM6SF2* polymorphisms by the odds ratio calculate, and to create a predictive model of advanced liver fibrosis. All statistical analyses were performed with R-commander v.2.6–2.

## 3. Results

### 3.1. Characteristics of the Study Cohort

A total of 140 patients fulfilling inclusion criteria were identified. Baseline characteristics are shown in Table 1.

T2D was present in 93 patients (66.4%). T2D treatment approaches included metformin (77%), either as a single treatment (27%) or co-administered with insulin (23%), Glucagon-Like Peptide-1 (GLP-1) analogues (9%), inhibitors of Sodium-glucose cotransporter-2 (iSGLT2) (12%), or inhibitors of Dipeptidyl Peptidase IV (iDPP-IV) (6%); insulin alone 9%, and diet only 14%.

Liver biopsy findings are shown in Table 2. No significant differences regarding the steatosis grade and NASH activity scored between T2D and non-T2D patients were found. Non-advanced fibrosis (stage 0–2) was present in 61 cases (44%), while 79 (56%) presented advanced fibrosis (stage 3–4). Patients with T2D showed a significantly higher proportion of advanced fibrosis than non-T2D patients (68% versus 34%, *p* < 0.001).

### 3.2. Genetic Analysis

The *PNPLA3* and *TM6SF2* genes were analyzed in all 140 NASH patients (Table 3). No deviation from the Hardy-Weinberg equilibrium for either the *PNPLA3* (*p* = 0.497) or *TM6SF2* (*p* = 0.081) genotypes was detected. The *PNPLA3* p.I148M minor allele was carried by 47% and the *TM6SF2* p.E167K minor allele was detected in 11% of NASH cohort. Of note, 24% of patients were homozygous carriers of the *PNPLA3* p.I148M allele, while we detected only 3% of homozygous for the *TM6SF2* p.E167K variant.

The frequency of *PNPLA3* genotypes carrying the minor allele (CG + GG) was 70%, while for the *TM6SF2* genotypes carrying the minor allele (CT + TT) it was 20%. No significant differences were found in the allelic distribution between non-T2D and T2D patients.

### 3.3. Risk Factors of Advanced Liver Fibrosis

Among the entire cohort, the risk factors associated with advanced liver fibrosis in the univariate analysis were age, T2D, HbA1c, the G allele for the *PNPLA3* p.I148M variant, the presence of T allele for the *TM6SF2* p.E167K variant, as well as the presence of either of two polymorphisms (G allele in *PNPLA3* p.I148M variant and T allele in *TM6SF2* p.E167K variant), as reflected in Table 4. T2D alone was a strong independent risk factor of advanced liver fibrosis in the NASH cohort (OR = 4.01; 95%CI 1.93–8.56; *p* < 0.001).

In the sub cohort of T2D patients, the presence of G allele for the *PNPLA3* p.I148M variant (OR = 2.57; 95%CI 1.03–6.41; *p* = 0.043) and also the presence of at least one of two polymorphisms were associated with advanced fibrosis (OR = 3.53; 95%CI 1.31–9.56; *p* = 0.013) (Table 5). When T2D was combined with each of the two polymorphisms (G allele in *PNPLA3* p.I148M variant or T allele in *TM6SF2* p.E167K variant), the OR significantly increased (OR = 14.69; 95%CI 3.03–77.35; *p* = 0.001 and OR = 11.45; 95%CI 3.16–41.55; *p* < 0.001, respectively).

A multivariate model was performed in the entire cohort, combining T2D (OR = 4.67; 95%CI 2.13–10.20; *p* < 0.001) and presence of two polymorphisms variables (OR = 3.94; 95%CI 1.63–9.54; *p* = 0.002), improving the diagnostic performance of the variables treated individually.

Regarding non-T2D patients, the univariate analysis showed that only age and HOMA-IR were found to be associated with advanced fibrosis (OR = 1.12; 95%CI 1.04–1.20; *p* = 0.004 and OR = 1.31; 95% CI 1.06–1.74; *p* = 0.034, respectively). The multivariate analysis (including age, HOMA-IR and the presence of one of the G allele in *PNPLA3* p.I148M variant or T allele in *TM6SF2* p.E167K variant) showed that the classical cut-off used for IR (3.02) did not significantly influence the risk of advanced fibrosis. Nevertheless, when a more severe IR degree (HOMA-IR ≥ 5.20) was included in the model, a significant association with the likelihood of advanced fibrosis was found Please confirm intended meaning has been retained), as shown in Table 5.

## 4. Discussion

In the present study we showed that *PNPLA3* and *TM6SF2* gene polymorphisms significantly increased the risk of advanced fibrosis in patients with NASH, while more importantly, IR increased the risk of advanced liver fibrosis in non-diabetic patients.

The demographic and metabolic characteristics of our cohort were similar to previous studies [28,29,30,31]. The severity of steatosis and NASH activity were similar between diabetic and non-diabetic patients, in accordance with previous studies [31]. Nevertheless, in our study, 64% of the patients presented mild steatosis and 56% advanced fibrosis in the histological analysis, a higher proportion than previously reported in similar cohorts [28,29,30,32]. One possible explanation is that our cohort is not based on general population, but on selected patients with a high suspicion of liver disease that were previously referred to a liver specialized clinic from a tertiary centre. In exchange, patients with T2D presented with higher proportion of advanced fibrosis than the non-diabetic patients, which is also consistent with prior reports [33,34,35]. Regarding the association between NAFLD and T2D, it has been shown that NASH represents the sole feature of liver damage in metabolic syndrome, driven mainly by insulin resistance [36].

*PNPLA3* p.I148M and *TM6SF2* p.E167K variants have been previously associated with NAFLD and NASH and advanced fibrosis [32,37,38]. In our cohort, the frequency of the minor (G) allele at *PNPLA3* rs738409 was higher (47%) than that reported in other studies where NAFLD was diagnosed by liver biopsy [39,40]. Furthermore, a global frequency of 21% for this allele variant is estimated [41]. By contrast, minor allele frequency (MAF) (T) at *TM6SF2* rs58542926 (11%) was similar to previous data (9%) in NAFLD patients [42].

*PNPLA3* p.I148M and *TM6SF2* p.E167K variants have been also associated to a higher risk of developing hepatic steatosis and advanced fibrosis [17,43,44,45]. Our results confirm those of previous studies, namely *PNPLA3* rs738409 and *TM6SF2* rs58542926 alone (OR = 2.20; 95%CI 1.06–4.64; p = 0.036 and OR = 2.79; 95%CI 1.15–7.57; *p* = 0.031, respectively) or in combination (OR = 3.25; 95%CI 1.44–7.65; *p* = 0.005) were associated with significant risk of advanced fibrosis in our cohort. This agrees with previous reports showing an additive effect of risk alleles accumulation on liver injury in NAFLD [15,20,30,42]. Mechanistically, a plausible explanation for the synergistic effect of the combination of the two alleles could be that the two polymorphisms are increasing the liver lipid content in different pathways. Specifically, *PNPLA3* polymorphism is responsible for the increase and accumulation of liver lipids and the *TM6SF2* polymorphism is reducing the exportation of liver lipids.

When the presence of T2D was added to each gene variant model, the risk of presenting advanced fibrosis significantly increased (OR = 14.69; 95%CI 3.03–77.35; *p* = 0.001 and OR = 11.45; 95%CI 3.16–41.55; *p* < 0.001, respectively). Opposite to previous studies that pointed out age to be a risk factor for advanced liver fibrosis [29,46,47], our study in T2D patients showed that age was not associated with an increased risk of advanced fibrosis, suggesting that T2D alone is a very strong risk factor, exceeding the risk of age in liver fibrosis.

Meanwhile, amongst non-diabetic patients, either *PNPLA3* rs738409 or *TM6SF2* rs58542926 or in combination were not associated to the risk of advanced liver fibrosis, whereas age and HOMAR-IR, as biomarker of hepatic IR, were indeed risk factors of advanced fibrosis in the absence of T2D diagnosis. Previous data in the literature have linked the IR with progression of fibrosis in patients with obesity and NASH [48,49]. Boursier et al. included HOMA-IR in an algorithm to identify fibrosis and proposed a cut-off HOMA-IR > 10 to predict worsening in patients with NASH [50]. Classical HOMA-IR cut-off ≥ 3.02 was described in Caucasian young populations where a subset of patients with obesity showed a new HOMA-IR cut-off ≥ 3.42 [26]. Obesity is a metabolic risk factor that predisposes to IR [51]. Patients in our cohort had a mean BMI of 32 kg/m^2^, while overall HOMA-IR mean value was 7.41, and 5.40 in non-T2D patients, defining significant IR regardless the cut-off for normality (either 3.02 or 3.42) [26].

The mechanistic relationship between the IR and the *PNPLA3* rs738409 and *TM6SF2* rs58542926 polymorphisms is incompletely explored and warrants further studies. Data so far associated the presence of both variants with hepatic triglyceride content [15,52]. Furthermore, the liver triglycerides deposition has been found to be more pronounced with a high-sucrose diet, which is well-known risk factor for IR [53,54,55]. Increased synthesis of liver triglycerides is due to an imbalance between de novo synthesis, reuptake of free fatty acids, and hepatic oxidation. In a context of IR, lipid oxidation decreases considerably, compared to lipid synthesis processes. Increased oxidative stress and mitochondrial dysfunction are related with the progression from steatosis to NASH. Hyperinsulinemia can cause a greater synthesis of VLDL particles in fasting, which in addition to a lowered liver secretion, favors the development of liver steatosis [56]. Recently, Luukkonen et al. [22], suggested that IR and genetic factors independently relate to NASH, and advanced fibrosis. Nevertheless, in their study no combined score was developed.

In order to investigate the interaction between the presence of the studied polymorphisms and the IR, several models were explored for creating a combined score that included age, *PNPLA3* rs738409 and *TM6SF2* rs58542926 polymorphisms and HOMA-IR. Interestingly, in the model that included the classical cut-off for IR for the Spanish population (HOMA-IR 3.02), the IR had no role in the risk of advanced fibrosis. In exchange, the third model that included the cut-off of HOMA-IR > 5.20 the presence of significant IR strongly predicted the risk of advanced fibrosis in the presence of age and the studied polymorphisms. We propose this new cut-off of HOMA-IR > 5.20 as biomarker of advanced liver fibrosis in non-T2D patients.

Our study has some limitations that should be noticed and restrict the extrapolation of our results to the general population, such as: (a) the sample size may have limited the interpretation of the role that T2D and *PNPLA3* p.I148M and *TM6SF2* p.E167K polymorphisms play in the development of advanced fibrosis; (b) T2D patients were predominant in the NASH cohort and may have disturbed the results regarding the effect of both polymorphisms in advanced fibrosis in the absence of diabetes. However, we consider this proportion as a reflection of real-life situations, being T2D the more frequent comorbidity in NAFLD patients; (c) finally, the low number of homozygote’s mutant alleles made necessary the use of a dominant genetic model; therefore, the effect of mutated homozygotes versus heterozygotes in the development of advanced fibrosis has not been fully verified.

In summary, we found that the presence of T2D or significant IR, defined by a new proposed cut-off of HOMA-IR > 5.20, on top of *PNPLA3* rs738409 and *TM6SF2* rs58542926 polymorphisms were associated to advanced liver fibrosis. Our results suggested that in patients with T2D and/or significant IR, a genetic study should be performed in order to identify those patients at higher risk of developing advanced liver fibrosis and center the efforts in a personalized follow-up and pharmacological treatments when available. Further studies are needed in order to validate and confirm our results, but it seems that IR mediates the risk of progression towards advanced liver fibrosis induced by *PNPLA3* rs738409 and *TM6SF2* rs58542926 polymorphisms.

## Figures and Tables

**Table 1 biomedicines-10-01015-t001:** Clinical characteristics of the study cohort and subdivided respect to T2D status.

Variable	Whole NASH Cohort (*n* = 140)	NASH Patients without T2D (*n* = 47)	NASH Patients with T2D (*n* = 93)	*p* Value
Age (years)	59 (10)	55 (12)	60 (9)	0.001
Female gender, *n* (%)	81 (58%)	25 (53%)	56 (60%)	0.542
BMI (kg/m^2^)	32 (5)	31 (5)	32 (5)	0.209
Waist circumference (cm)	108 (12)	105 (13)	109 (11)	0.120
Fasting glucose (mg/dL)	129 (55)	94 (12)	147 (60)	<0.001
HbA1c (%)	6.5 (1.4)	5.5 (0.4)	7.1 (1.5)	<0.001
HOMA-IR	7.41 (6.40)	5.40 (2.99)	8.69 (6.08)	0.007
Triglycerides (mg/dL)	153 (113–206)	136 (114–170)	161 (113–216)	0.088
Cholesterol LDL (mg/dL)	116 (37)	130 (33)	109 (37)	0.001
Cholesterol HDL (mg/dL)	49 (12)	52 (12)	47 (11)	0.036
ALT (UI/L)	46 (31–71)	55 (34–103)	45 (30–63)	0.059
AST (UI/L)	42 (29–59)	44 (30–59)	41 (27–59)	0.615
GGT (UI/L)	73 (41–160)	58 (40–150)	74 (41–166)	0.681

Values are mean (standard deviation), number (%), or median (Q1–Q3). BMI: body mass index; HOMA-IR: Homeostasis model assessment of insulin resistance; ALT: Alanine aminotransferase; AST: Aspartate aminotransferase; GGT: Gamma-glutamyl transpeptidase. *p* values are obtained between “without T2D” and “with T2D” group’s comparison.

**Table 2 biomedicines-10-01015-t002:** Grade of steatosis; NASH activity score and fibrosis in the entire NASH cohort and subdivided respect to T2D status.

Biopsy Results	Whole NASH Cohort (*n* = 140)	NASH Patients without T2D (*n* = 47)	NASH Patients with T2D (*n* = 93)	*p* Value
Steatosis				
1	90 (64)	32 (68)	58 (62)	0.782
2	39 (28)	12 (25)	27 (29)	
3	11 (8)	3 (6)	8 (9)	
NASH activity score				
≤3	55 (39)	19 (40)	36 (38)	0.554
4	40 (29)	15 (32)	25 (27)	
5	27 (19)	6 (13)	21 (23)	
≥6	18 (13)	7 (15)	11 (12)	
Fibrosis ranges				
0–2	61 (44)	31 (66)	30 (32)	<0.001
3–4	79 (56)	16 (34)	63 (68)	

Values are number (%). *p* values are obtained between “without T2D” and “with T2D” group’s comparison.

**Table 3 biomedicines-10-01015-t003:** Allelic distribution in the entire NASH cohort and subdivided respect to T2D status.

Variable		Whole NASH Cohort (*n* = 140)	NASH Patients without T2D (*n* = 47)	NASH Patients with T2D (*n* = 93)	*p* Value
*PNPLA3*	CC	42 (30)	12 (26)	30 (32)	0.532
	CG + GG	98 (70)	35 (74)	63 (68)	
*TM6SF2*	CC	112 (80)	41 (87)	71 (76)	0.195
	CT + TT	28 (20)	6 (13)	22 (24)	

Values are number (%). *PNPLA3*: patatin-like phospholipase domain containing protein 3; *TM6SF2*: transmembrane 6 superfamily member 2. *p* values are obtained between “without T2D” and “with T2D” group’s comparison.

**Table 4 biomedicines-10-01015-t004:** Risk factors for developing advanced liver fibrosis in the entire NASH cohort and subdivided respect to T2D status.

	Whole NASH Cohort (*n* = 140)				NASH Patients without T2D (*n* = 47)				NASH Patients with T2D (*n* = 93)			
Factor	OR	95% CI	*p* Value	AUC	OR	95% CI	*p* Value	AUC	OR	95% CI	*p* Value	AUC
Univariate analysis												
Age	1.07	1.03–1.11	0.001	0.67	1.12	1.04–1.20	0.004	0.78	1.02	0.97–1.08	0.387	0.56
Sex	0.92	0.47–1.81	0.807	0.51	1.78	0.52–6.10	0.356	0.57	0.54	0.21–1.35	0.186	0.57
Presence of T2D	4.01	1.93–8.56	<0.001	0.65	-	-	-	-	-	-	-	-
HOMA-IR	1.08	1.01–1.18	0.053	0.68	1.31	1.06–1.74	0.034	0.71	1.01	0.95–1.10	0.727	0.56
HbA1C	1.38	1.05–1.82	0.022	0.62	1.37	0.24–7.77	0.726	0.51	1.07	0.79–1.45	0.675	0.51
BMI	0.98	0.92–1.05	0.634	0.53	1.04	0.92–1.17	0.538	0.55	0.93	0.86–1.02	0.125	0.61
*PNPLA3* p.I148M	2.20	1.06–4.64	0.036	0.58	3.33	0.63–17.57	0.156	0.60	2.57	1.03–6.41	0.043	0.61
*TM6SF2* p.E167K	2.79	1.15–7.57	0.031	0.58	2.15	0.38–12.15	0.385	0.55	2.60	0.79–8.52	0.114	0.58
*PNPLA3* p.I148M and *TM6SF2* p.E167K *	3.25	1.44–7.65	0.005	0.60	6.14	0.70–53.24	0.101	0.61	3.53	1.31–9.56	0.013	0.62
T2D+*PNPLA3* p.I148M	14.69	3.03–77.35	0.001	0.67	-	-	-	-	-	-	-	-
T2D+*TM6SF2* p.E167K	11.45	3.16–41.55	<0.001	0.75	-	-	-	-	-	-	-	-
Multivariate analysis												
Presence of T2D	4.67	2.13–10.20	<0.001	0.71								
*PNPLA3* p.I148M and *TM6SF2* p.E167K *	3.94	1.63–9.54	0.002									

* At least one of the two polymorphisms.

**Table 5 biomedicines-10-01015-t005:** Multivariate analysis in non-diabetic NASH patients for developing advanced liver fibrosis.

		OR	95%CI	*p* Value	AUC	Sensitivity (%)	Specificity (%)
Model 1	Age	1.11	1.03–1.23	0.017	0.89	80	83
	HOMA-IR	1.56	1.09–2.50	0.033			
	*PNPLA3* p.I148M and *TM6SF2* p.E167K *	31.29	2.16–2163.53	0.042			
Model 2	Age	1.11	1.03–1.22	0.013	0.85	87	83
	HOMA-IR ≥ 3.02	2.79	0.48–22.43	0.275			
	*PNPLA3* p.I148M and *TM6SF2* p.E167K *	8.77	1.12–193.81	0.074			
Model 3	Age	1.14	1.05–1.26	0.004	0.89	80	83
	HOMA-IR ≥ 5.20	14.33	2.14–18.66	0.014			
	*PNPLA3* p.I148M and *TM6SF2* p.E167K *	19.04	1.71–650.84	0.042			

* At least one of the two polymorphisms.

## Data Availability

The data that support the findings of this article are available from the corresponding authors on reasonable request.

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
