# Peer review of "Influence of Type 2 Diabetes in the Association of PNPLA3 rs738409 and TM6SF2 rs58542926 Polymorphisms in NASH Advanced Liver Fibrosis"

_biomedicines, 2022, doi:10.3390/biomedicines10051015_

Round 1
Reviewer 1 Report
I enjoyed reviewing this interesting paper. The study is methodologically correct and well written. The conclusions are supported by the results.
This reviewer only suggests enriching the manuscript with some interesting recent references closely related to this study.
1- Notably, it was recently observed by liver biopsy that steatohepatitis represents the sole feature of liver damage in type 2 diabetes (PLoS One. 2017 Jun 1;12(6):e0178473. doi: 10.1371/journal.pone.0178473.). This observation confirms the authors' hypothesis that T2D and IR status increase the risk of advanced fibrosis. This important issue should be commented on in the discussion, and the above reference should be added.
2- A couple of updated reviews have recently well described how PNPLA3 plays a crucial role in both the development and progression of NAFLD (Antioxidants (Basel). 2021 Feb 10;10(2):270. doi: 10.3390/antiox10020270.) and how, on this basis, NAFLD represents an important cardiovascular risk factor (Rev Cardiovasc Med. 2021 Sep 24;22(3):755-768. doi: 10.31083/j.rcm2203082). These issues should be added in the introduction as well as previous references.
Author Response
We thank the reviewer for his kind comments regarding our manuscript (biomedicines-1680247) tittled “Influence of type 2 diabetes in the association of PNPLA3 rs738409 and TM6SF2 rs58542926 polymorphisms in NASH advanced liver fibrosis”.
This reviewer only suggests enriching the manuscript with some interesting recent references closely related to this study.
1- Notably, it was recently observed by liver biopsy that steatohepatitis represents the sole feature of liver damage in type 2 diabetes (PLoS One. 2017 Jun 1;12(6):e0178473. doi: 10.1371/journal.pone.0178473.). This observation confirms the authors' hypothesis that T2D and IR status increase the risk of advanced fibrosis. This important issue should be commented on in the discussion, and the above reference should be added.
As requested we have added the suggested reference and changed the text accordingly in the new version of the manuscript as shown below:
“Regarding the association between NAFLD and T2D, it has been shown that NASH rep-resents the sole feature of liver damage in metabolic syndrome, driven mainly by insulin resistance [36]”.
2- A couple of updated reviews have recently well described how PNPLA3 plays a crucial role in both the development and progression of NAFLD (Antioxidants (Basel). 2021 Feb 10;10(2):270. doi: 10.3390/antiox10020270.) and how, on this basis, NAFLD represents an important cardiovascular risk factor (Rev Cardiovasc Med. 2021 Sep 24;22(3):755-768. doi: 10.31083/j.rcm2203082). These issues should be added in the introduction as well as previous references.
As requested we have added the two references suggested by this reviewer in the Introduction (Pag2 line 49 and line 64) of the revised manuscript.
Reviewer 2 Report
Manuscript ID: biomedicines-1680247
Type of manuscript: Article
Title: Influence of type 2 diabetes in the association of PNPLA3 rs738409 and TM6SF2 rs58542926 polymorphisms in NASH advanced liver fibrosis
Background:
Since there are no reports linking PNPLA3 and TM6SF2 gene variants with T2D and IR in the pathophysiology of NASH and in advanced liver fibrosis. The current study aimed to investigate the interaction between PNPLA3 and TM6SF2 gene variants, in T2D and IR with advanced fibrosis in a cohort of patients with NASH. Their results showed that IR mediated advanced liver fibrosis progression which was induced by PNPLA3 rs738409 and TM6SF2 rs58542926 polymorphisms.
This study sheds light on the importance of screening genetically patients with T2D/IR for PNPLA3 rs738409 and TM6SF2 rs58542926 polymorphisms. Since these patients are more prone to developing advanced liver fibrosis, aiming for personalized pharmacological treatments.
Comments to Authors:
It is a very well written study and present important findings pertaining to the importance of this area of research.
- Elaborate a bit in the introduction on: “PNPLA3 rs738409 and TM6SF2 rs58542926 polymorphisms”
- Make sure that the percentages in line 225: “our study 64% of the patients presented steatosis and 56% advanced fibrosis” are reported accurately.
- Can we conclude that there is a kind of interaction between those two variants or each one can act independently? Meaning is there any additive/synergistic effect? If so how could that explained mechanistically (you can elaborate on this in the discussion section).
This paper can be accepted after addressing the minor revision above.
Author Response
We thank the reviewer for his kind comments, corrections and suggestions regarding our manuscript (biomedicines-1680247) tittled “Influence of type 2 diabetes in the association of PNPLA3 rs738409 and TM6SF2 rs58542926 polymorphisms in NASH advanced liver fibrosis”.
Comments to the authors:
Elaborate a bit in the introduction on: “PNPLA3 rs738409 and TM6SF2 rs58542926 polymorphisms”.
As requested we have added a paragraph regarding PNPLA3 and TM6SF2 polymorphisms in the Introduction section of the revised manuscript as shown below:
“PNPLA3 or adiponutrin participates in intracellular lipid remodeling and the rs738409 polymorphism is associated with increased hepatocellular triglyceride accumulation by restricting substrate access to the enzyme’s catalytic site. On the other hand, TM6SF2 plays a role in VLDL export from liver to serum, where SNP rs58542926 causes impairment in the lipid exportation and is associated with elevations in serum aspartate aminotransferase (AST) and alanine aminotransferase (ALT) [21].”
Make sure that the percentages in line 225: “our study 64% of the patients presented steatosis and 56% advanced fibrosis” are reported accurately.
We have revised this issue and clarified in the new version of the manuscript as shown below:
“Nevertheless, in our study 64% of the patients presented mild steatosis and 56% advanced fibrosis in the histological analysis, a higher proportion than previously reported in similar cohorts [28-30,32].”
Can we conclude that there is a kind of interaction between those two variants or each one can act independently? Meaning is there any additive/synergistic effect? If so how could that explained mechanistically (you can elaborate on this in the discussion section).
We thank the reviewer for raising this point and indeed there is a synergistic effect of these two polymorphisms and we have speculated about the possible mechanism involved in the new version of the manuscript as shown below:
“This is in agreement with previous reports showing an additive effect of risk alleles ac-cumulation on liver injury in NAFLD [15,20,30,42]. Mechanistically, a plausible explanation for the synergistic effect of the combination of the two alleles could be that the two polymorphisms are increasing the liver lipid content in different pathways. Specifically, PNPLA3 polymorphism is responsible for the increase and accumulation of liver lipids and the TM6SF2 polymorphism is reducing the exportation of liver lipids”.